# A rapid review of COVID-19's global impact on breast cancer screening participation rates and volumes from January to December 2020

**Reagan Lee[1], Wei Xu[1,2], Marshall Dozier[3], Ruth McQuillan[1], Evropi Theodoratou[2], Jonine Figueroa[2,4]\*, On Behalf of UNCOVER and the International Partnership for Resilience in CancerSystems (I-PaRCS), Breast Cancer Working Group 2**

[1]Usher Institute, University of Edinburgh, Edinburgh, United Kingdom; [2]Centre for Global Health, University of Edinburgh, Edinburgh, United Kingdom; [3]Information Services, University of Edinburgh, Edinburgh, United Kingdom; [4]Division of Cancer Epidemiology and Genetics, National Cancer Institute, Bethesda, United States

**\*For correspondence:**
jonine.figueroa@ed.ac.uk

**Competing interest:** The authors declare that no competing interests exist.

**Abstract** COVID-19 has strained population breast mammography screening programs that aim to diagnose and treat breast cancers earlier. As the pandemic has affected countries differently, we aimed to quantify changes in breast screening volume and uptake during the first year of COVID-19 . We systematically searched Medline, the World Health Organization (WHO) COVID-19 database, and governmental databases. Studies covering January 2020 to March 2022 were included. We extracted and analyzed data regarding study methodology, screening volume, and uptake. To assess for risk of bias, we used the Joanna Briggs Institute (JBI) Critical Appraisal Tool. Twenty-six cross-sectional descriptive studies (focusing on 13 countries/nations) were included out of 935 independent records. Reductions in screening volume and uptake rates were observed among eight countries. Changes in screening participation volume in five nations with national population-based screening ranged from –13 to –31%. Among two countries with limited population-based programs, the decline ranged from –61 to –41%. Within the USA, population participation volumes varied ranging from +18 to –39%, with suggestion of differences by insurance status (HMO, Medicare, and low-income programs). Almost all studies had high risk of bias due to insufficient statistical analysis and confounding factors. The extent of COVID-19-induced reduction in breast screening participation volume differed by region and data suggested potential differences by healthcare setting (e.g., national health insurance vs. private healthcare). Recovery efforts should monitor access to screening and early diagnosis to determine whether prevention services need strengthening to increase the coverage of disadvantaged groups and reduce disparities.

## Editor's evaluation

This study presents important evidence of the impact of the covid pandemic on breast cancer screening globally but with important variations by healthcare setting. The data analysis is comprehensive, using solid systematic review methods. The results will be of interest to public health policymakers and health care and cancer control practitioners and researchers across the globe.

**Figure 1.** PRISMA Flow Diagram for Record Identification, Screening and Inclusion for Analysis (*Page et al., 2021*).

## Introduction

Breast cancer is the most common cancer worldwide, with 2.3 million cases diagnosed and 685,000 deaths in 2020 (*WHO, 2021*). Mammography-based screening programs allow for early detection of breast cancers for earlier intervention and disease stage that improves patient outcomes (*IARC, 2022*). Early detection and diagnosis from screening may reduce mortality by up to 65% among breast cancer patients (*Berry et al., 2005*). Populations with a good uptake rate in screening programs can achieve a 90% 5-year survival rate in patients who received an early diagnosis attributed to screening (*WHO, 2021*).

COVID-19 affected global health systems and has strained population breast mammography screening programs. Previous work on modeled evaluations and a focus on tumor staging and mortality as outcomes suggested that scenarios are likely to differ by region and organization of delivery of breast cancer screening (*Figueroa et al., 2021*). In different countries, screening models vary from population-based to opportunistic screening (offered to patients in healthcare settings – more common in private healthcare) (*IARC, 2016*).

Here we aimed to quantify systematically breast screening participation rates before and after the first COVID-19 wave amidst the suspensions in nations with/without opportunistic screening programs. This was performed by investigating two primary study outcomes: changes in screening volume and participation uptake rates.

## Results

*Figure 1* summarizes the search strategy. The initial search retrieved 1207 articles and 935 independent records. After screening (see 'Methods'), 26 cross-sectional studies from 13 countries were eligible for inclusion (*Table 1*). We counted Scotland and England as two separate national entities due to the devolved healthcare systems. However, it should be noted that breast screening policies and practice between NHS Scotland and NHS England are similar. In total, 7 reports came from Europe (*Campbell et al., 2021*; *Jidkova et al., 2022*; *Knoll et al., 2022*; *Eijkelboom et al., 2021*; *Losurdo et al., 2022*; *Toss et al., 2021*; *NHS England, 2021*), 2 from Oceania (*BreastScreen Australia, 2020*; *BreastScreen Aoteroa, 2022*), 1 from Asia (*Shen et al., 2022*), 2 from South America (*Bessa, 2021*; *Ribeiro et al., 2022*), and 14 from North America (*Chiarelli et al., 2021*;

**Table 1.** Descriptive characteristics of included cross-sectional studies (n = 26).

| Study | Publication type | Study design | Country | Region (If not national) | Total Female Population of Study Area | Sample size | Study screening data source | Screening (National/Regional) | Screening age range | Screening type | Screening time comparison | Types of Restrictions present over study period† | | | | | | | | COVID-19 7 day new infection rate in region of focus (per 100000)* | |
|---|---|---|---|---|---|---|---|---|---|---|---|---|---|---|---|---|---|---|---|---|---|
| | | | | | | | | | | | | International Travel Limits | Internal Movement Controls | Stay at home requirement | Public transport closure | Ban on gatherings of >10 people | Public events ban | Workplace closure | School closure | Minimum infection rate in study period | Maximum infection rate in study period |
| **Europe (n=7)** | | | | | | | | | | | | | | | | | | | | | |
| **Campbell et al., 2021** | Peer-reviewed | Cross sectional | Scotland (UK) | | 2728000 | Not specified | NHS Scotland | National | 50–70 | Digital Mammography | Aug – Dec 2019 vs Aug –Dec 2020 | Yes | Yes | No | No | Yes | Yes | Yes | No | 10.14 | 212.67 |
| **Jidkova et al., 2022** | Peer-reviewed | Cross sectional | Belgium | Flanders | 3382265 | Not specified | Flanders Online Screening Database | Regional | 50–69 | Digital Mammography | Jul – Nov 2019 vs Jul – Nov 2020 | Yes | Yes | Yes | No | Yes | Yes | Yes | Yes | 3.58 | 580.63 |
| **Knoll et al., 2022** | Preprint | Cross sectional | Austria | Innsbruck | 567300 | 596 | Database from gynecological oncological center in Austria, Tyrol | Local | 45–69 years invited for screening. Women aged 40–44 years and 70–75 years may opt in | Digital Mammography | Mar – Dec 2019 vs Mar – Dec 2020 | Yes | Yes | Yes | No | Yes | Yes | Yes | Yes | no data | no data |
| **Eijkelboom et al., 2021** | Peer-reviewed | Cross sectional | Netherlands | | 8701000 | 3371 | Netherlands Cancer Registry | National | 50–75 | Digital Mammography | Jan – Feb 2020 vs Jul – Aug 2020 | Yes | No | No | No | Yes | No | Yes | No | 0.32 | 67.25 |
| **Losurdo et al., 2022** | Peer-reviewed | Cross sectional | Italy | Friuli Venezia Giulia | 624418 | 58643 | "Data-Breast" database of the "Eusoma certified SSD Breast Unit of Trieste and from the Surgical Department of DAI Chirurgia Generale—ASUGI. | Regional | 50–69 | Digital Mammography | Oct – Dec 2019 vs Oct – Dec 2020 | Yes‡ | Yes‡ | Yes‡ | No‡ | Yes‡ | Yes‡ | Yes‡ | Yes‡ | 19.2 | 497.6 |
| **Toss et al., 2021** | Peer-reviewed | Cross sectional | Italy | Northern Italy, Emilia Romagna | 2291000 | 24994 | Emilia Romagna National Healthcare System | Regional | 45–79 | Digital Mammography | 2019 vs 2020 | Yes‡ | Yes‡ | Yes‡ | Yes‡ | Yes‡ | Yes‡ | Yes‡ | Yes‡ | 4.00 | 390.9 |
| **NHS England, 2021** | Government paper | Cross sectional | England (UK) | | 33940000 | 2230000 | NHS England | National | 50–71 | Digital Mammography | 2019 vs 2020 | Yes | Yes | Yes | Yes | Yes | Yes | Yes | Yes | 0.00 | 92.36 |
| **Oceania (n=2)** | | | | | | | | | | | | | | | | | | | | | |
| **BreastScreen Australia, 2020** | Government Paper | Cross sectional | Australia | | 12780000 | Not specified | BreastScreen Australia | National | 50–74 | Digital Mammography | May – Sep 2018 vs May – Sep 2020 | Yes | Yes | Yes | No | Yes | Yes | Yes | Yes | 0.18 | 13.31 |
| **BreastScreen Aotearoa, 2022** | Government Paper | Cross sectional | New Zealand | | 2497000 | Not specified | BreastScreen Aotearoa | National | 45–69 | Digital Mammography | May – Dec 2018 vs May – Dec 2020 | Yes | Yes | Yes | No | Yes | Yes | Yes | Yes | 0 | 1.06 |
| **Asia (n=1)** | | | | | | | | | | | | | | | | | | | | | |
| **Shen et al., 2022** | Peer-reviewed | Cross sectional | China | Taiwan | 11981657 | 699911 | Taiwan National Infectious Disease Statistics system | Regional | 40–69 | Digital Mammography | Jan – Apr 2019 vs Jan – Apr 2020 | Yes | No | No | No | No | No | No | Yes | no data | no data |
| **Americas (n=16)** | | | | | | | | | | | | | | | | | | | | | |

*Table 1 continued on next page*

Table 1 continued

| Study | Publication type | Study design | Country | Region (if not national) | Total Female Population of Study Area | Sample size | Study screening data source | Screening (National/Regional) | Screening age range | Screening type | Screening time comparison | International Travel Limits | Internal Movement Controls | Stay at home requirement | Public transport closure | Ban on gatherings of >10 people | Public events ban | Workplace closure | School closure | Minimum infection rate in study period | Maximum infection rate in study period |
|---|---|---|---|---|---|---|---|---|---|---|---|---|---|---|---|---|---|---|---|---|---|
| | | | | | | | | | | | | | Types of Restrictions present over study period† | | | | | | | COVID-19 7 day new infection rate in region of focus (per 100000)* | |
| Bessa, 2021 | Peer-reviewed | Cross sectional | Brazil | | 106500000 | (2019: 20636636; 2020: 211409758) | Brazilian Unified Health System (SUS) | National | 50–69 | Digital Mammography | 2019 vs 2020 | Yes | Yes | Yes | Yes | Yes | Yes | Yes | Yes | 0.00 | 149.68 |
| Ribeiro et al., 2022 | Peer-reviewed | Cross sectional | Brazil | | 106500000 | 5996798 | Brazilian National Health Service (SUS) Outpatient Information System (SIA/SUS), SUS Hospital Information System (SIH/SUS), High Complexity Procedure Authorizations database (APAC), Cancer Information System (ISCAN). | National | 50–69 | Digital Mammography | Jul – Dec 2019 vs Jul – Dec 2020 | Yes | Yes | Yes | Yes | Yes | Yes | Yes | Yes | 53.72 | 149.68 |
| Chiarelli et al., 2021 | Peer-reviewed | Cross sectional | Canada | Ontario | 7371000 | 426967 | Ontario Breast Screening Program (OBSP) | Regional | 50–74 | Digital Mammography, MRI (High risk) | Jul - Dec 2019 vs Jul - Dec 2020 | Yes | Yes | Yes | No | Yes | Yes | Yes | Yes | 3.99 | 117.01 |
| Walker et al., 2021 | Peer-reviewed | Cross sectional | Canada | Ontario | 7371000 | 605889 (2019) 284242 (2020) | Ontario Breast Screening Program (OBSP) | Regional | 50–74 | Digital Mammography | Modelled 2019 data vs Dec 2020 | Yes | Yes | Yes | No | Yes | Yes | Yes | Yes | 75.74 | 117.01 |
| Doubova et al., 2021 | Peer-reviewed | Cross sectional | Mexico | | 64570000 | 1431216 | Mexican Institute of Social Security (IMSS) | National | 40 - unspecified | Digital Mammography | Jan 2019 – Mar 2020 vs Apr – Dec 2020 | Yes | Yes | Yes | No | No | Yes | Yes | Yes | 2.60 | 61.12 |
| Chen et al., 2021 | Peer-reviewed | Cross sectional | USA | | 167500000 | Not specified | HealthCore Integrated Research Database | National | 50–79 years | Digital Mammography | Jul 2019 vs Jul 2020 | Yes | Yes | Yes | No | Yes | Yes | Yes | Yes | 119.03 | 142.00 |
| Amornsiripanitch et al., 2021 | Peer-reviewed | Cross sectional | USA | Massachusetts | 3537000 | 32387 | Electronic medical record (Epic, Verona, WI) - Massachusetts. One tertiary care academic center, a community hospital, a specialized cancer center, three outpatient imaging centers, one urban healthcare center, and one mobile mammography van | Regional | 40 - unspecified years | Digital Mammography | Jun – Aug 2019 vs Jun – Aug 2020 | Yes | Yes | No | No | Yes | Yes | No | No | 17.06 | 53.09 |

Table 1 continued

| Study | Publication type | Study design | Country | Region (If not national) | Total Female Population of Study Area | Sample size | Study screening data source | Screening (National/ Regional) | Screening age range | Screening type | Screening time comparison | Types of Restrictions present over study period | | | | | | | | COVID-19 7 day new infection rate in region of focus (per 100000) | |
|---|---|---|---|---|---|---|---|---|---|---|---|---|---|---|---|---|---|---|---|---|---|
| | | | | | | | | | | | | International Travel Limits | Internal Movement Controls | Stay at home requirement | Public transport closure | Ban on gatherings of >10 people | Public events ban | Workplace closure | School closure | Minimum infection rate in study period | Maximum infection rate in study period |
| Becker et al., 2021 | Peer-reviewed | Cross sectional | USA | Michigan | 5062000 | 7250080 | Women enrolled in Health Managed Organization (HMO) Blue Cross Blue Shield (BCBS) in Michigan | Regional | 40–74 | Digital Mammography | Dec 2019 vs Dec 2020 | Yes | Yes | Yes | No | Yes | Yes | Yes | Yes | 147.56 | 328.94 |
| DeGroff et al., 2021 | Peer-reviewed | Cross sectional | USA | | 167500000 | 630264 | Breast and Cervical Cancer Early Detection Program (NBCCEDP) Database, which provides cancer screening services to women with low income and inadequate health insurance | National | 40–74 | Digital Mammography | Jun 2019 vs Jun 2020 | Yes | Yes | Yes | No | Yes | Yes | Yes | Yes | 45.46 | 103.84 |
| Dennis et al., 2021 | Peer-reviewed | Cross sectional | USA | | 167500000 | 475083 | Behavioral Risk Factor Surveillance System (BRFSS) survey database | National | 40–74 | Digital Mammography | 2014–2019 vs 2020 | Yes | Yes | Yes | Yes | Yes | Yes | Yes | Yes | 0.00 | 460.68 |
| Fedewa et al., 2021 | Peer-reviewed | Cross sectional | USA | | 167500000 | 2019: 142003 2020: 150630 | Data from 32 CHCs of the American Cancer Society's Community Health Advocates Implementing Nationwide Grants for Empowerment and Equity (CHANGE) grant program to increase BCSRs and follow-up care | National | 50–74 | Digital Mammography | 2019 vs 2020 | Yes | Yes | Yes | Yes | Yes | Yes | Yes | Yes | 0.00 | 460.68 |
| Lehman et al., 2022 | Preprint | Cross sectional | USA | | 167500000 | 29276 | Screening database over 5 facilities | National | Unspecified | Digital Mammography | 2019 vs 2020 | Yes | Yes | Yes | No | Yes | Yes | Yes | Yes | 0.00 | 460.68 |
| London et al., 2022 | Peer-reviewed | Cross sectional | USA | | 167500000 | 34000000 (full study including colorectal cancers) | TriNetX Research Network | National | Unspecified | Digital Mammography | Jul – Dec 2019 vs Jul – Dec 2020 | Yes | Yes | Yes | No | Yes | Yes | Yes | Yes | 74.54 | 460.68 |
| Miller et al., 2021 | Peer-reviewed | Cross sectional | USA | Virginia | 2757460 | Not specified | Institution Database, University of Virginia | Regional | Unspecified (45 - 70) | Digital Mammography | Jan – Nov 2019 vs Jan - Nov 2020 | Yes | Yes | Yes | No | Yes | Yes | Yes | Yes | no data | no data |

Table 1 continued on next page

*Table 1 continued*

| Study | Publication type | Study design | Country | Region (If not national) | Total Female Population of Study Area | Sample size | Study screening data source | Screening (National/ Regional) | Screening age range | Screening type | Screening time comparison | Types of Restrictions present over study period† | | | | | | | | COVID-19 7 day new infection rate in region of focus (per 100000)* | |
| | | | | | | | | | | | | International Travel Limits | Internal Movement Controls | Stay at home requirement | Public transport closure | Ban on gatherings of >10 people | Public events ban | Workplace closure | School closure | Minimum infection rate in study period | Maximum infection rate in study period |
|---|---|---|---|---|---|---|---|---|---|---|---|---|---|---|---|---|---|---|---|---|---|
| **Sprague et al., 2021** | Peer-reviewed | Cross sectional | USA | | 167500000 | 461083 | 62 radiology facilities of Breast Cancer Surveillance Consortium | National | 40–79 | Digital Mammography | Jan–Jul 2019 vs Jan–Jul 2020 | Yes | Yes | Yes | No | Yes | Yes | Yes | Yes | 0.00 | 142.00 |
| **Nyante et al., 2021** | Peer-reviewed | Cross sectional | USA | North Carolina | 5099371 | 42412 | 7 academic and community breast imaging facilities in North Carolina | Regional | 40–79 | Digital Mammography | Modelled Sep 2019 data vs Sep 2020 | Yes | No | No | No | No | Yes | Yes | Yes | 80.27 | 91.26 |

England's and Scotland's NHS systems are devolved and, therefore, are separate national entities. However, they hold similar screening criterion where breast screening policy in the NHS (across the UK) is that all women aged 50–70 y + 364 d are invited for breast screening once every 3 y.

*These infection rates were region-specific and analogous to the region the study involved. If study period was <1 mo, only infection data from the first and last week of the period will be collected. If study period was over the whole year of 2020, the earliest available public health data was used (e.g., study period started from January 2020 but data was only available in March, March data used as first interval of analysis). It should be noted that there is reporting bias here as testing rates may differ between countries. These infection incidence rates were based on national/regional data depending on whether the study population originated from an entire nation or a limited region within a nation. (*Dipartimento della Protezione Civile, 2023a; Dipartimento della Protezione Civile, 2023b; Government of Ontario, 2023; Government of the Netherlands, 2023; Cooper et al., 2023; IARC, 2022; Mathieu, 2022; MDHHS, 2023; Medicaid.gov, 2022; NCDHHS, 2023; NHS England, 2021; OECD, 2021a; PAHO, 2020; Sciensano, the Belgian Institute for Health, 2023; SPICe, 2023; State of Michigan, 2020; State of North Carolina, 2020; The Scottish Government, 2022; UK Government, 2023; WHO, 2023; WHO, 2022; WHO, 2022; WHO, 2022; Worldometer, 2022; WHO, 2021; Yucatan Times, 2021).*

†Types of restrictions will include restrictions that were withdrawn at any point of the study period. Restrictions present were classified as per non-pharmacological interventions mentioned by the paper Li et al., 2021 (*The Temporal Association of introducing and lifting non-pharmaceutical interventions with the time-varying reproduction number (R) of SARS-COV-2: A modelling study across 131 countries*, The Lancet Infectious Diseases, if restrictions were introduced/withdrawn during the study period, it will still be indicated as a 'Yes'. Data from Oxford COVID-19 policy tracker; devolved state-wide healthcare organization websites in Canada, the USA, and UK was used to assess this.

‡Data was unavailable for regions in this country, national restrictions were assessed instead.

*Walker et al., 2021*; *Doubova et al., 2021*; *Chen et al., 2021*; *Amornsiripanitch et al., 2021*; *Becker et al., 2021*; *DeGroff et al., 2021*; *Dennis et al., 2021*; *Fedewa et al., 2021*; *Lehman et al., 2022*; *London et al., 2022*; *Miller et al., 2021*; *Sprague et al., 2021*; *Nyante et al., 2021*). The most frequently reported country was the USA (n = 11). Studies examined either regional (n = 13) or national populations (n = 13).

During COVID-19, many countries implemented various mitigation methods to reduce transmission and of course mortality. To summarize these different infection control measures, *Table 1* shows that all 13 countries/nations had international movement controls in place, 23 study-specific regions had internal movement controls, 21 study-specific regions had stay-at home requirements in place, 1 study-specific region (Northern Italy, Emilia Romagna) had public transport closures, 23 study-specific regions had bans on gatherings >10 people, 24 study-specific regions had public events bans in place, 24 study-specific regions had workplace closures in place, and 23 study-specific regions had in-person school closures in place (*Mathieu, 2022*; *CIHI, 2022*; *Commonwealth of Massachusetts, 2021*; *Commonwealth of Virginia, 2023*; *Cooper et al., 2023*; *SPICe, 2023*; *State of Michigan, 2020*; *State of North Carolina, 2020*).

Analysis of data from all studies was limited from January 1, 2020, to December 31, 2020.

## Screening volume changes over study period

Summary data from 17 studies in eight countries reporting breast cancer screening volumes, and data from 106,484,908 women before and after COVID-19 infection control measures were extracted (data from 2017 to 2020 were the comparison time period, *Table 2*; *Doubova et al., 2021*; *Bessa, 2021*; *Ribeiro et al., 2022*; *Chiarelli et al., 2021*; *Losurdo et al., 2022*; *Walker et al., 2021*; *NHS England, 2021*; *Shen et al., 2022*; *BreastScreen Australia, 2020*; *DeGroff et al., 2021*; *Lehman et al., 2022*; *Amornsiripanitch et al., 2021*; *Sprague et al., 2021*; *London et al., 2022*; *Miller et al., 2021*; *Nyante et al., 2021*; *Becker et al., 2021*). Most studies that showed calendar period trends of screening volume noted temporal variation with declines especially at the height of the pandemic between March and May 2020. In countries with national screening programs, a negative change in screening volume was reported, with the lowest volume change estimated at –12.86% in Australia (*BreastScreen Australia, 2020*), followed by –15.80% in England (*NHS England, 2021*). A larger negative change in screening volume was observed in Brazil (–41.49%) (*Ribeiro et al., 2022*) and Mexico (–61.30%) (*Doubova et al., 2021*). It should be noted that Brazil and Mexico have a lower proportion of population-based breast screening coverage relative to other countries; Brazil having coverage of ~24% and Mexico having ~20% coverage of the eligible population (*OECD, 2021a*; *Unger-Saldaña et al., 2020*). A significant proportion of breast screening in Brazil and Mexico consists of opportunistic screening programs.

In the USA, which has mix of insurance providers there was a wide range of change in screening volume. Using data from Health Managed Organization (HMO) Blue Cross Blue Shield (BCBS) from the state of Michigan, the authors observed temporal changes in rates with an increase slightly above 2019 levels in the last few months of 2020, with an 18.10% overall increase in screening volume (*Becker et al., 2021*). Although rates were above 2019 levels, the authors noted that the odds that a woman received breast cancer screening remained 20% lower in 2020 relative to 2019 (*Becker et al., 2021*). This was consistent with the decrease in screening volume that was generally observed from six studies with data among populations wholly or partially covered by national insurance (*Lehman et al., 2022*; *Amornsiripanitch et al., 2021*; *Sprague et al., 2021*; *London et al., 2022*; *Miller et al., 2021*; *Nyante et al., 2021*). Percentage decreases ranged from –36.50 (*Lehman et al., 2022*) to –9.80% (*Miller et al., 2021*). Data from the USA National Breast and Cervical Cancer Early Detection Program (NBCCEDP), which provides cancer screening services to women with low income and inadequate health insurance, reported a greater decrease (–39.00%) in volume (*DeGroff et al., 2021*). Two other studies had smaller populations with less certainty and wider confidence intervals, with one reporting an 8% increase (*Nyante et al., 2021*) and the other a –10% decline (*London et al., 2022*). In the USA, where there is a mix of national (Medicare) and private insurance depending on age, screening volume changes were similar to other national screening programs at –36.50% (*Lehman et al., 2022*). In contrast, a positive increase in volume was observed among private insurance providers +30% (*London et al., 2022*).

**Table 2.** Breast cancer screening volumes change among 106,484,908 subjects from eight countries.

Percentage change in volume of breast cancer screening (N = 17)

| Study | Country | Region | National/regional (scope of study population*) | Type of breast screening program employed within the study population | Sample size | Screening timeframe comparison | Volume change relative to non-COVID-19 period (%) |
|---|---|---|---|---|---|---|---|
| *Europe (n = 2)* | | | | | | | |
| **Losurdo et al., 2022** | Italy | Friuli Venezia Giulia | Regional | Population-based screening present in country | 58,643 | Oct–Dec 2019 vs. Oct–Dec 2020 | 11.90 |
| **NHS England, 2021** | UK | England | National | Population-based screening present in country | 3,420,000 | Monthly average 2019 vs. monthly average 2020 | 15.80 |
| *Oceania (n = 1)* | | | | | | | |
| **BreastScreen Australia, 2020** | Australia | NA | National | Population-based screening present in country | 802,146 | May–Sep 2018 vs. May–Sep 2020 | 12.88 |
| *Asia (n = 1)* | | | | | | | |
| **Shen et al., 2022** | China | Taiwan | Regional | Population-based screening present in country | 699,911 | Jan–Apr 2019 vs. Jan–Apr 2020 | 22.07 |
| *America (n = 13)* | | | | | | | |
| **Bessa, 2021** | Brazil | NA | National | Population-based screening present in country† | (2019: 20,636,636; 2020: 21,140,958) | 2019 vs. 2020 | 42.72 |
| **Ribeiro et al., 2022** | Brazil | NA | National | Population-based screening present in country but private sector databases included Brazilian National Health Service (SUS), Outpatient Information System (SIA/SUS), SUS Hospital Information System (SIH/SUS), High Complexity Procedure Authorizations database (APAC), Cancer Information System (ISCAN) | 5,996,798 | Jul–Dec 2019 vs. Jul–Dec 2020 | 41.49 |
| **Doubova et al., 2021** | Mexico | NA | National | Population-based screening present in country ‡ | 1,431,216 | Jan 2019–Mar 2020 vs. Apr–Dec 2020 | 61.30 |
| **Chiarelli et al., 2021** | Canada | Ontario | Regional | Population-based screening present in country | 426,967 | Jul–Dec 2019 vs. Jul–Dec 2020 | 31.30 |
| **Walker et al., 2021** | Canada | Ontario | Regional | Population-based screening present in country | 890,131 | Modeled 2019 data vs. Dec 2020 | 22.80 |
| **Lehman et al., 2022** | USA | NA | National | Privatized system with mix of national and private insurance usage | 29,276 | 2019 vs. 2020 | 36.50 |
| **Miller et al., 2021** | USA | North Carolina | Regional | Privatized system with mix of national and private insurance usage | 8,536,000 | Jan–Nov 2019 vs. Jan–Nov 2020 | 9.80 |
| **Amornsiripanitch et al., 2021** | USA | Massachusetts | Regional | Privatized system with mix of national and private insurance usage | 32,387 | Jun–Aug 2019 vs. Jun–Aug 2020 | 10.50 |
| **London et al., 2022** | USA | NA | National | Privatized system with mix of national and private insurance usage | 34,000,000 | Dec 2019 vs. Dec 2020 | 20.00 |
| **DeGroff et al., 2021** | USA | NA | National | The National Breast and Cervical Cancer Early Detection Program (NBCCEDP) that provides cancer screening services to women with low income and inadequate health insurance | 630,264 | Jun 2019 vs. Jun 2020 | 39.00 |

*Table 2 continued on next page*

*Table 2 continued*

**Percentage change in volume of breast cancer screening (N = 17)**

| Study | Country | Region | National/ regional (scope of study population*) | Type of breast screening program employed within the study population | Sample size | Screening timeframe comparison | Volume change relative to non-COVID-19 period (%) |
|---|---|---|---|---|---|---|---|
| *Becker et al., 2021* | USA | Michigan | Regional | Health Managed Care Organization (HMO)-based screening (database covers HMO data from Michigan) | 7,250,080 | Dec 2019 vs. Dec 2020 | 18.10 |
| *Sprague et al., 2021* | USA | NA | National | Privatized system with mix of national and private insurance usage | 461,083 | Jul 2019 vs. Jul 2020 | 10.30 (–20.40 to 6.60) |
| *Nyante et al., 2021* | USA | North Carolina | Regional | Privatized system with mix of national and private insurance usage | 42,412 | Modeled Sep 2019 data vs. Sep 2020 | 9.00 |

NA indicates not applicable. For studies conducted in the USA, ACS guidelines were used as the data collection comparator starting point where March–June 2020 was considered to be a suspension in screening.

*This column highlights the origin of the study population in which whether it was drawn from a specific region within a nation, or if the study population was drawn from the entire country.

†The study population from this specific study (***Bessa, 2021***) was solely drawn from a national population-based screening database in Brazil. It should be noted that Brazil has a lower proportion of population-based breast screening coverage relative to other countries; having a coverage of 24% in the eligible population (***Unger-Saldaña et al., 2020***).

‡It should be noted that Mexico has a lower proportion of population-based breast screening coverage relative to other countries due to recent introduction; having ~20% coverage of the eligible population (***OECD, 2021b***; ***PAHO, 2020***).

## Screening participation uptake rate changes

A total of nine cross-sectional studies reported breast cancer screening participation rates and represented >46,257,402 participants from varying calendar periods across five countries (***Amornsiripanitch et al., 2021***; ***Dennis et al., 2021***; ***Fedewa et al., 2021***; ***Chen et al., 2021***; ***NHS England, 2021***; ***Campbell et al., 2021***; ***Bessa, 2021***; ***BreastScreen Aoteroa, 2022***; ***Jidkova et al., 2022***). There was considerable variability in change (***Table 3***), ranging from +2–8% in Scotland to –43.54% in Brazil (***Campbell et al., 2021***; ***Bessa, 2021***). In the USA, there was a consistent negative change in screening participation uptake rates (***Amornsiripanitch et al., 2021***; ***Dennis et al., 2021***; ***Fedewa et al., 2021***; ***Chen et al., 2021***).

## Study quality

The quality of the included studies was assessed using the JBI tool (***Table 4***). A weakness across most studies was failure to identify and consider confounding factors. From ***Table 4***, 25 studies had no issues defining the inclusion sample. Nineteen studies were clear in defining the study setting and subjects. Studies had no issues quantifying exposure of COVID-19, although this was based on temporality since all healthcare systems globally were affected (***Worldometer, 2022***). All studies apart from ***Becker et al., 2021*** had no issue measuring the condition through either screening appointment attendance or insurance claims data. Most studies (65%, N = 17) did not define confounding factors regarding measurement of primary outcomes. Regarding comparison of volumes of screening prior to COVID-19 and observed periods, these studies did not provide source of reduction in screening capacity (e.g., due to social distancing or participation uptake). Twenty-three studies failed to provide strategies to address confounding factors (e.g., elucidating reduction in capacity and presenting it as a proportion to overall volume).

Four studies (***Bessa, 2021***; ***Becker et al., 2021***; ***London et al., 2022***; ***Doubova et al., 2021***) had unclear reasons for selection of study subjects and control groups (***London et al., 2022***), confounding factors that were not indicated, nor strategies included to solve this. Among these four papers, vague definition of control groups resulted in a poor comparator, resulting in unreliable outcome measures.

Twenty-three studies provided basic statistical analyses (e.g., mean, adjusted rates per population) with basic data presentation. Statistical analyses were not performed in three government papers (***BreastScreen Australia, 2020***; ***NHS England, 2021***; ***BreastScreen Aoteroa, 2022***). Twenty-two studies were unclear or did not provide sufficient descriptive statistical analyses regarding comparison of control data to observed data. Statistical analyses were performed in four studies. This includes

**Table 3.** Breast cancer screening participation uptake rates change from nine studies from five countries.

**Percentage change in participation uptake rate of breast cancer screening (N = 9)**

| Study | Country | Region | National/regional (scope of study population)* | Type of breast screening program employed within the study population | Sample size | Screening timeframe comparison | Participation rate change relative to non-COVID-19 period |
|---|---|---|---|---|---|---|---|
| *Europe (n = 3)* | | | | | | | |
| **NHS England, 2021** | UK | England | National | Population-based screening available in country | 3,420,000 | 2019 vs. 2020 | 11.80% |
| **Campbell et al., 2021** | UK | Scotland | National | Population-based screening available in country | NA | Aug–Dec 2019 vs. Aug–Dec 2020 ‡ | +10.96% (Aug 2020); +2–8% (Sep 2020–Mar 2021 vs. Sep 2019– Mar 2020)‡ |
| **Jidkova et al., 2022** | Belgium | Flanders | Regional | Population-based screening available in country | NA | Jul–Dec 2019 vs. Jul–Dec 2020 | 1.0% (–1.3; –0.7) |
| *Oceania (n = 1)* | | | | | | | |
| **BreastScreen Aoteroa, 2022** | New Zealand | NA | National | Population-based screening available in country | NA | Dec 2018/2019 vs. May–Dec 2020 | 6.70% |
| *Americas (n = 5)* | | | | | | | |
| **Bessa, 2021** | Brazil | NA | National | Population-based screening available in country† | (2019: 20,636,636; 2020: 21,140,958) | 2019 vs. 2020 | 43.54% |
| **Dennis et al., 2021** | USA | NA | National | Privatized system with mix of national and private insurance usage | 475,083 (age: 50–74) 117,498 (age: 40–49) | 2014–2019 vs. 2020 | 5.30% (50–79); 7.20% (40–49) |
| **Fedewa et al., 2021** | USA | NA | National | Privatized system with mix of national and private insurance usage | 434,840 | 2019 vs. 2020 | 8.00% |
| **Amornsiripanitch et al., 2021** | USA | Massachusetts | Regional | Privatized system with mix of national and private insurance usage | 32,387 | Jun–Aug 2019 vs. Jun–Aug 2020 | 14.80% |
| **Chen et al., 2021** | USA | NA | National | Privatized system with mix of national and private insurance usage | NA | Jul 2019 vs. Jul 2020 | 3.33% |

NA indicates not applicable For studies conducted in the USA, ACS guidelines were used as the data collection comparator starting point where Mar-Jun 2020 was considered to be a suspension in screening.

*This column highlights the origin of the study population in which whether it was drawn from a specific region within a nation, or if the study population was drawn from the entire country.

†The study population from this specific study (**Bessa, 2021**) was solely drawn from a national population-based screening database in Brazil. It should be noted that Brazil has a lower proportion of population-based breast screening coverage relative to other countries; having a coverage of 24% in the eligible population (**Unger-Saldaña et al., 2020**).

‡It should be noted that this study presented a range of values (2–8%) comparing the uptake rate from Sep 2020 to Mar 2021 vs. Sep 2019 to Mar 2020. As the timeframe of Jan–Mar 2021 was not within the scope of the study, we used the point estimate of the uptake rate in Aug 2020 vs. Aug 2019 as our last available data point instead.

**Table 4.** Summary of results of appraisal of all included studies with Joanna Briggs Institute (JBI) Critical Appraisal Tool for cross-sectional studies.

JBI Critical Appraisal Tool for cross-sectional studies appraisal table

| Study | Were the criteria for inclusion in the sample clearly defined? | Were the study subjects and the setting described in detail? | Was the exposure measured in a valid and reliable way? | Were objective, standard criteria used for measurement of the condition? | Were confounding factors identified? | Were strategies to deal with confounding factors stated? | Were the outcomes measured in a valid and reliable way? | Was appropriate statistical analysis used? |
|---|---|---|---|---|---|---|---|---|
| Amornsiripanitch et al., 2021 | Y | Y | Y | Y | Y | N | Unclear | Unclear |
| Becker et al., 2021 | Y | Y | Y | N | Y | N | N | Unclear |
| Bessa, 2021 | Y | Unclear | Y | Y | N | N | N | Unclear |
| Campbell et al., 2021 | Y | Unclear | Y | Y | Unclear | N | Y | Unclear |
| Chen et al., 2021 | Y | Unclear | Y | Y | Y | N | Y | Unclear |
| Chiarelli et al., 2021 | Y | Y | Y | Y | Unclear | N | Y | Unclear |
| DeGroff et al., 2021 | Y | Y | Y | Y | Y | N | Y | Unclear |
| Dennis et al., 2021 | Y | Y | Y | Y | N | N | Y | Unclear |
| Doubova et al., 2021 | Y | Unclear | Y | Y | N | N | N | Y |
| Jidkova et al., 2022 | Y | Y | Y | Y | Unclear | N | Y | Unclear |
| Knoll et al., 2022 | Y | Y | Y | Y | N | N | Y | Unclear |
| Fedewa et al., 2021 | Y | Y | Y | Y | N | N | Y | Unclear |
| BreastScreen Australia, 2020 | Y | Y | Y | Y | N | N | Y | N |
| Eijkelboom et al., 2021 | Y | Y | Y | Y | Y | Y | Y | Unclear |
| Lehman et al., 2022 | N | N | Y | Y | Y | N | Y | Unclear |
| London et al., 2022 | N | N | Y | Y | N | N | N | Unclear |
| Losurdo et al., 2022 | Y | Y | Y | Y | N | N | Y | Unclear |
| Walker et al., 2021 | Y | Y | Y | Y | Unclear | N | Y | Unclear |
| Toss et al., 2021 | Y | Y | Y | Y | N | N | Y | Unclear |
| Shen et al., 2022 | Y | Y | Y | Y | Unclear | N | Y | Unclear |
| Ribeiro et al., 2022 | Y | Y | Y | Y | N | N | Y | Unclear |
| Miller et al., 2021 | Y | Unclear | Y | Y | Y | Y | Y | Y |
| Sprague et al., 2021 | Y | Y | Y | Y | Unclear | Y | Y | Y |
| Nyante et al., 2021 | Y | Y | Y | Y | Y | Y | Y | Y |
| NHS England, 2021 | Y | Y | Y | Y | N | N | Y | N |

*Table 4 continued*

**JBI Critical Appraisal Tool for cross-sectional studies appraisal table**

| Study | Were the criteria for inclusion in the sample clearly defined? | Were the study subjects and the setting described in detail? | Was the exposure measured in a valid and reliable way? | Were objective, standard criteria used for measurement of the condition? | Were confounding factors identified? | Were strategies to deal with confounding factors stated? | Were the outcomes measured in a valid and reliable way? | Was appropriate statistical analysis used? |
|---|---|---|---|---|---|---|---|---|
| *BreastScreen Aoteroa, 2022* | Y | Y | Y | Y | N | N | Y | N |

Green = yes; yellow = unclear; orange = no.

provision of odds ratios by *Doubova et al., 2021* and *Miller et al., 2021*, Poisson estimation of a 95% confidence interval (95% CI) by *Sprague et al., 2021*, and 95% confidence intervals from comparison of means from *Nyante et al., 2021*.

## Discussion

We previously reported on modeled evaluations that estimated short- and long-term outcomes for various scenarios and changes in breast screening volume, uptake rates, and breast cancer diagnosis rates (*Figueroa et al., 2021*; *WHO, 2021*). In this rapid review, we show that during COVID-19 there was a generally reported reduction in breast cancer screening volume and participation uptake rate that varied by healthcare setting (e.g., national population-based screening vs. opportunistic or private healthcare). Our data suggests that volume and participation uptake are important metrics that requires monitoring by health systems and could inform prevention and early diagnosis efforts, especially if certain groups are not participating.

Non-pharmaceutical interventions were essential and effective in containing the spread of COVID-19 in the era without vaccines; these extend to domestic/international movement controls, social distancing, and ban on events and gatherings and workplace/school closure (*Li et al., 2021*; *Talic et al., 2021*). While these measures were important to reduce the mortality directly related to COVID-19, they also had indirect effects on other health services including breast cancer screening. In this rapid review, we provide evidence that screening volume and participation uptake rates were reduced but this reduction varied by region and healthcare system.

In a systematic review and meta-analysis, data from 72 studies were used to investigate the effectiveness of public health measures in reducing COVID-19 incidence and transmission (*Talic et al., 2021*). The meta-analysis pooled an estimate from eight studies and indicated that handwashing (Relative Risk (RR): 0.47; 95% CI: 0.19–1.12), mask-wearing (RR: 0.47; 95% CI: 0.29–0.75), and physical distancing (RR: 0.75; 95% CI: 0.59–0.95) were associated with the reduction in COVID-19 incidence. The remaining public health measures including quarantine and isolation, universal lockdowns, and closures of borders, schools, and workplaces which could not be included in the meta-analysis were evaluated in a narrative way. The findings validated the effectiveness of both individual and packages of public health measures on the transmission of SARS-CoV-2 and incidence of COVID-19. However, the majority of included studies had moderate risk of bias based on quality assessment. For breast cancer screening, the importance of mitigation measures that emphasized physical distancing to have been the most important in reducing screening, both for general population participation but also at healthcare facilities aiming to reduce transmission (*Figueroa et al., 2021*).

Reductions in screening capacity due to physical distancing are likely another source for screening volume reductions. Screening capacity reductions were caused by social distancing, staggered appointments, staff exposure to COVID-19, and cleaning measures. This likely resulted in reductions in time allocated for screening to occur (*Walker et al., 2021*; *Sprague et al., 2021*). *Sprague et al., 2021* considered screening capacity when assessing screening volume. Even though screening capacity recovered to pre-pandemic levels in July 2020, screening volume experienced a 10.8% decrease relative to the control period. Reductions in screening capacity were potentially not the sole factor to screening volume reductions. However, most publications included in our rapid review did not collect data regarding screening capacity, so we cannot determine the proportion of change in screening volume that was attributed to either reduction in screening capacity or change in patient willingness to attend screening. Future analyses are needed where both measures are obtained, which would inform what measures are needed (e.g., information campaigns to alleviate patient fears or increase clinical staffing for catch-up of missed appointments).

Our data supports differences by healthcare system that were particularly evident in data from the USA where there is a mix of private and national healthcare (Medicare) for persons 65+ [https://www.medicare.gov/]. *DeGroff et al., 2021*, who studied populations reliant solely on national health insurance, showed larger screening volume reductions (–39.00%). This was relative to studies focusing solely on populations with private insurances or studies including patients from both groups (–36.50 to +30%). *Amornsiripanitch et al., 2021*, which included national and private insurance patients, corroborate this. Medicaid and Medicare patients had –17.06% screening volume reduction compared to –10.50% experienced by the entire population. *Miller et al., 2021* suggest that opportunity cost of attending breast screening in lower income groups (e.g., employment) may have led to decreased

breast screening in such populations. Some literature showed increases in screening volumes (*Nyante et al., 2021*; *Becker et al., 2021*) and uptake rates (*Campbell et al., 2021*). Increased volume (+9%) from *Nyante et al., 2021* could be inconclusive as the observed screening volume was compared against a modeled 2019 population that was used to simulate a 2020 population in the absence of COVID-19. Although this study was robust, limited data collection till September 2020 did not show full extent of change regarding screening volumes after lifting of COVID-19 suspension guidelines in June 2020. From trends explored in study, breast screening rates were possibly recovering in the study population (USA) in late 2020, but more data is required. The Affordable Care Act may have alleviated breast screening cost through health insurance coverage reforms (*Zhao et al., 2020*). However, this does not address other underlying socioeconomic inequalities (e.g., high cost of treatment, time off from work due to sickness). Patients from deprived backgrounds may be fearful of dealing with the consequences of abnormal screening results (e.g., treatment). This may strain patient finances worsened by COVID-19, potentially explaining lower screening volumes and uptake. Future data on patient characteristics including insurance status, socioeconomic, and race/ethnicity could inform targeted campaigns to reduce inequities if disparities exist.

*Becker et al., 2021* showed a screening volume increase after the lifting of COVID-19 suspension guidelines. This study focused on patients who utilize solely private insurance. Patients already paying for services may be more inclined to maximize utilization of coverage. However, this study states that the odds that a woman received breast cancer screening remained 20% lower in 2020 (OR = 0.80 [0.80, 0.81]) relative to 2019. This study scored poorly in the JBI appraisal tool due to poor outcome measurement; it was unclear how the odds ratio was derived, therefore, increasing the risk of bias of this study. Unusual outcome measures were used, that being the claims invoice for the service. This appeared unreliable; it was unclear whether paying for the service equates to a fulfilled appointment. Invoices could be delayed, making it unclear when the screening took place. This study's evidence quality needs to be increased for results to be conclusive.

*Campbell et al., 2021* state a 10.96% increase in uptake rate in Scotland. This study population (within the study period) solely included patients who had their appointments cancelled in March 2020 due to the first lockdown and high-risk patients. This particular patient group may have an increased urgency to catch up on screening. This could have contributed to the increased uptake rate of screening in Scotland in the study period. The increase in uptake rates could also be attributed to the increased accessibility for patients due to the 'work-from-home' model and increased health consciousness due to COVID-19. Neither raw data nor sample size was defined in the study and will require future analysis.

Due to the inherent weaknesses of a rapid review, certain limitations are present within the study as explored below. However, this study can be expanded upon by various means (also explored below) to further elucidate the global impact of COVID-19 on breast cancer detection and subsequent care. Other limitations include COVID-19 context as an evolving field with fast publication turnovers; more papers could have been published since the review started. This issue could be partially addressed by completing a repeat search with employment of forward and backward citation tracking, while including more gray literature sources apart from governmental databases (e.g., private screening databases). Other limitations included studies had insufficient data for combined analysis regarding COVID-19 waves past December 2020. Additionally, the data obtained was cross-sectional instead of cohort-based; we were unable to analyze trends and recovery in breast cancer screening rates and incidence rates over time. Exclusion of non-English-language literature was a weakness. Many countries with extensive population-based breast screening programs that were affected by COVID-19 in Europe and Asia were unaccounted for; the inclusion of additional data would be useful to clarify the impact of the pandemic on breast cancer screening program uptake. Furthermore, it should also be noted that COVID-19 infection rates were not reported by the included studies and data from governmental/health board websites may not report study-specific region infection rates.

In summary, screening volume and uptake rates were generally reduced but many studies showed gains over time even if overall a decline in screening volume was observed. These declines were likely due to the first COVID-19 wave where many healthcare facilities paused non-essential services. Volume and uptake reductions of smaller magnitudes were observed, and our data suggest some difference depending on region and healthcare coverage. Access to screening services may increase marginalization of some vulnerable groups in the USA due to the pandemic, and recovery efforts

to reduce disparities in access to screening and early diagnosis should be monitored to determine whether prevention services need strengthening. Participation uptake and volume are not conclusive endpoints themselves, and future work from registries and other data sources are needed to determine whether there has been any impact on incidence, stage, and mortality outcomes.

## Methods

We performed a rapid review (*Tricco et al., 2015*), where systematic review processes were modified to facilitate project completion within a shortened timeframe. Searches were limited to two databases and English-language governmental gray literature.

### Literature search

RL ran a systematic search on 'Ovid MEDLINE(R) and In-Process, In-Data-Review & Other Non-Indexed Citations' Database and WHO COVID-19 Literature Database, with entry date limits from January 1, 2020, to March 12, 2022. In brief, we performed the search with MeSH subject headers and free text terms for 'COVID-19,' 'Breast Neoplasms,' and 'Mass screening.' Our search strategies are listed in *Table 5*. We searched gray literature from government health websites known to have data from population-based screening programs. These consisted of the National Cancer Institute (USA), CDC (USA), NHS (National Healthcare Service) UK database, BreastScreen Australia, and BreastScreen Aotearoa New Zealand. We further screened reference lists of the retrieved eligible publications to identify additional relevant studies. An English-language restriction was placed on the searches. Deduplication was carried out as part of upload to Covidence systematic review software, Veritas Health Innovation, Melbourne, Australia. Available at https://www.covidence.org/.

### Inclusion and exclusion criteria

The Population, Interventions, Comparator, Outcomes, and Study Characteristics (PICOS) model (*Schardt et al., 2007*) was used to determine eligibility criteria. A pilot literature screen (n = 10) was performed by RL with guidance from MD and JF to confirm validity of criteria. The population of focus are women eligible for breast cancer screening programs globally (population-based or opportunistic) or breast screening programs that are a part of the International Screening Cancer Network (ISCN). The intervention investigated involves the introduction of COVID-19 infection control measures. These were assumed to be present globally due to worldwide prevalence of COVID-19 by March 2020, chosen due to the WHO's declaration of a pandemic. We also added data on infection control measures based on *Li et al., 2021* 'The Temporal Association of introducing and lifting non-pharmaceutical interventions with the time-varying reproduction number (R) of SARS-COV-2: A modelling study across 131 countries', *The Lancet Infectious Diseases*, (see 'Data extraction' section for more detail). The comparator involved breast cancer screening statistics after COVID-19-related screening shutdown versus an analogous period in the previous years (e.g., comparing statistics in Australia from May to Sep 2020 against data from May to Sep 2018/2019) or any relevant period.

Outcomes assessed were the percentage change in 'volume' of breast screening participation, defined as total number of breast screening procedures; the percentage change in participation 'uptake rate' of breast screening program, defined as the percentage of the eligible population who attend screening; and incidence of breast cancer diagnosis. These were obtained through direct data extraction or calculated with data derived from the comparison of values from each comparator period. Full-text, English-language primary papers or governmental published gray literature were included. Studies with data entirely pertaining to diagnostic imaging were excluded or with future modeled data were excluded. All studies focused on women. Studies were required to have data on breast screening following the resumption of breast screening in countries with a screening shutdown.

### Title, abstract, full-text screen

Two reviewers (RL, JF) parallelly independently reviewed titles, abstracts, and subsequently full texts based on predefined inclusion and exclusion criteria. Deduplication of articles and screening was performed on Covidence. Conflict resolution was performed by discussion.

### Data extraction

Data extraction for each article was conducted by a single reviewer (RL). A second reviewer (WX) then checked for eligibility of extracted data in 70% of the texts. Any conflicts were resolved by

**Table 5.** Search strategies for rapid review of breast cancer participation and volume during COVID.

**Search string for Ovid MEDLINE(R) and In-Process, In-Data-Review & Other Non-Indexed Citations**

| Search number | Search domain | Search string in: [mp = title, book title, abstract, original title, name of substance word, subject heading word, floating subheading word, keyword heading word, organism supplementary concept word, protocol supplementary concept word, rare disease supplementary concept word, unique identifier, synonyms] |
|---|---|---|
| #1 | COVID-19 | (COVID-19 OR 2019 novel coronavirus disease OR 2019 novel coronavirus infection OR 2019 ncov disease OR 2019 ncov infection OR 2019-ncov disease OR 2019-ncov diseases OR 2019-ncov infection OR 2019-ncov infections OR covid 19 OR covid 19 pandemic OR covid 19 virus disease OR covid 19 virus infection OR covid-19 OR covid-19 pandemic OR covid-19 pandemics OR covid-19 virus disease OR covid-19 virus diseases OR covid-19 virus infection OR covid-19 virus infections OR covid19 OR coronavirus disease 19 OR coronavirus disease-19 OR disease 2019, coronavirus OR sars cov 2 infection OR sars coronavirus 2 infection OR sars-cov-2 infection OR sars-cov-2 infections OR severe acute respiratory syndrome coronavirus 2 infection OR disease, 2019-ncov OR disease, covid-19 virus OR infection, 2019-ncov OR infection, covid-19 virus OR infection, sars-cov-2 OR pandemic, covid-19 OR virus disease, covid-19 OR virus infection, covid-19 OR Coronavirus, 2019 Novel OR ncov OR covid* OR coronavirus* OR SARS* OR severe acute respiratory syndrome OR coronavirus pandemic OR coronavirus disease pandemic) |
| #2 | Breast cancer | (Breast Neoplasms OR Breast Carcinoma In Situ OR Carcinoma, Ductal, Breast OR Carcinoma, Lobular OR breast cancer OR breast carcinoma* OR breast malignant neoplasm* OR breast malignant tumo?r* OR breast neoplasm* OR breast tumo?r* OR cancer of breast? OR cancer of the breast? OR mammary carcinoma* OR mammary neoplasm* OR malignant neoplasm? of breast OR malignant tumo?r? of breast OR mammary cancer* OR neoplasm?, breast OR tumo?r, breast OR tumo?rs, breast OR cancer?, breast OR cancer?, mammary OR carcinoma?, human mammary OR carcinoma?, breast OR neoplasm?, human mammary OR breast carcinoma in situ OR lobular carcinoma in situ OR lcis, lobular carcinoma in situ OR mammary ductal carcinoma? OR carcinoma, ductal, breast OR carcinoma, infiltrating duct OR carcinoma, invasive ductal, breast OR carcinoma, mammary ductal OR carcinomas, infiltrating duct OR carcinomas, mammary ductal OR invasive ductal carcinoma, breast OR lobular carcinoma? OR carcinoma?, lobular OR breast* OR breast tumo?r OR breast tumo?rs OR breast malignant tumo?rs OR breast malignan* OR mammary malignan* OR malignant tumo?rs of breast OR neoplasm? of breast OR breast neoplasm OR lcis) |
| #3 | Mass screening | (Mass Screening OR Mass Chest X-ray OR Early Diagnosis OR Early Detection of Cancer OR Mammography OR screening* OR Ultrasonography, Mammary OR Ultrasonography OR mass chest x ray OR mass chest x-ray* OR mass chest xray* OR x-ray, mass chest OR x-rays, mass chest OR xray, mass chest OR xrays, mass chest OR disease early detection OR early detection of disease OR early diagnosis OR diagnosis, early OR cancer early detection OR cancer early diagnosis OR early detection of cancer OR early diagnosis of cancer OR digital breast tomosyntheses OR digital breast tomosynthesis OR x ray breast tomosynthesis OR x-ray breast tomosyntheses OR x-ray breast tomosynthesis OR breast tomosyntheses, digital OR breast tomosyntheses, x-ray OR breast tomosynthesis, digital OR breast tomosynthesis, x-ray OR breast tissue imaging OR mastography OR mass breast xray OR mass breast x-ray OR chest xray OR chest x-ray OR mammogra* OR program* OR ultrasonic* OR echograph* OR echotomograph* OR sonography* OR ultrasonograph* OR ultrasound* OR exam*) |
| #4 | Search string | 1 AND 2 AND 3 |
| #5 | Final search string | Limit 4 to English language |

**Search string for WHO COVID-19 Literature Database (updated to March 12, 2022)**

| Search number | Search concept | Title, abstract, subject |
|---|---|---|
| #1 | Breast cancer | ((Breast Neoplasms) OR (Breast Carcinoma In Situ) OR (Carcinoma, Ductal, Breast) OR (Carcinoma, Lobular) OR (breast cancer*) OR (breast carcinoma*) OR (breast malignant neoplasm*) OR (breast malignant tumo?r*) OR (breast neoplasm*) OR (breast tumo?r*) OR (cancer of breast?) OR (cancer of the breast?) OR (mammary carcinoma*) OR (mammary neoplasm*) OR (malignant neoplasm? of breast) OR (malignant tumo?r? of breast) OR (mammary cancer*) OR (breast carcinoma in situ) OR (lobular carcinoma in situ) OR (mammary ductal carcinoma*) OR (breast ductal carcinoma*) OR (infiltrating duct carcinoma*) OR (invasive ductal carcinoma) OR (mammary ductal carcinoma*) OR (invasive ductal breast carcinoma) OR (lobular carcinoma*) OR (breast tumo?r*) OR (breast malignant tumo?r*) OR (breast malignan*) OR (mammary malignan*) OR (malignant tumo?rs of breast*) OR (neoplasm? of breast) OR (lcis*)) |
| #2 | Screening | ((Mass Screening) OR (Mass Chest X-ray) OR (Early Diagnosis) OR (Early Detection of Cancer) OR (Mammography) OR (Ultrasonography, Mammary) OR (Ultrasonography) OR (national screening) OR (screening*) OR (mass chest x ray) OR (mass chest x-ray*) OR (mass chest xray*) OR (mass chest x-ray*) OR (disease early detection) OR (early detection of disease) OR (early diagnosis) OR (diagnosis, early) OR (cancer early detection) OR (cancer early diagnosis) OR (early detection of cancer) OR (early diagnosis of cancer) OR (digital breast tomosyntheses) OR (digital breast tomosynthesis) OR (x ray breast tomosynthesis) OR (x-ray breast tomosyntheses) OR (breast tomosynthesis*) OR (breast tomosynthesis*) OR (breast tissue imaging) OR (mastography) OR (mass breast xray) OR (mass breast x-ray) OR (chest xray) OR (chest x-ray) OR (mammogra*) OR (program*) OR (ultrasonic*) OR (echograph*) OR (ultrasonographic*) OR (sonography*) OR (echotomograph*) OR (ultrasound*) OR (exam*)) |
| #3 | Final search string | #1 AND #2 |
| | | English-language filter |

a third reviewer (JF). Data relevant to the evidence for population-based or opportunistic breast cancer screening programs during COVID-19 were extracted including citation details, publication type, study design, country, region, population, study setting, screening sample size, screening time-frame, screening volumes change (before/after COVID-19 infection control guidelines), screening participation uptake rates change (before/after COVID-19 infection control guidelines), and breast cancer incidence rates. A standardized data extraction form was created and piloted for extraction of primary outcome measures. Data pertaining to the presence of COVID-19 infection control measures and COVID-19 infection rates within the study region were also collected. We used the categories of infection control measures as presented in *Li et al., 2021* 'The Temporal Association of intro-ducing and lifting non-pharmaceutical interventions with the time-varying reproduction number (R) of SARS-COV-2: A modelling study across 131 countries.' In addition, we pulled data on infection rates collected from the Oxford COVID-19 policy tracker and devolved statewide healthcare organization websites in Canada, the USA, and the UK (*Mathieu, 2022*; *CIHI, 2022*; *Commonwealth of Massachusetts, 2021*; *Commonwealth of Virginia, 2023*; *Cooper et al., 2023*; *SPICe, 2023*; *State of Michigan, 2020*; *State of North Carolina, 2020*).

COVID-19 infection rates were defined as the incidence of COVID-19 cases within the area of focus per 100,000 people over 7 d (Formula = (Number of new cases within population over 7 days/ Total estimated population number) × 100,000). This was collected from the WHO COVID-19 Dash-board and various devolved health agencies of specific regions (*WHO, 2023*; *UK Government, 2023*; *Dipartimento della Protezione Civile, 2023a*; *Government of Ontario, 2023*; *Government of the Netherlands, 2023*; *MDHHS, 2023*; *MDHHS, 2023*; *The Scottish Government, 2022*).

### Risk-of-bias assessment

All studies included had cross-sectional designs. We used the JBI Critical Appraisal Tool for cross-sectional studies to assess the risk of bias of each article (*Critical appraisal tools, 2022*). The JBI checklist is available in *Table 4*. The risk of bias for each article was assessed by a single reviewer [RL], and a second reviewer [WX] cross-assessed the results and verified all related judgment and ratio-nales. Discrepancies were resolved through discussion and a joint reassessment of studies.

### Data synthesis

Data were synthesized descriptively since a meta-analysis was not appropriate due to heterogeneity of data. Data was collected by comparing outcome measures before and after COVID-19 infection control measures were introduced; this was presumed due to the worldwide prevalence of COVID-19 by March 2020.

Data were obtained from any point after lifting of COVID-19 breast screening suspension measures until an endpoint of December 31, 2020. If quantitative data was limited or if raw data was unavail-able, the last data point of the study was analyzed. This was compared to data from an analogous pre-COVID-19 period in 2018–2019, or if data was unavailable, against any relevant pre-pandemic period. For countries with no breast screening suspension in 2020, data from during COVID-19 was compared with an analogous period of 2018–2019. This phenomenon only occurred in Taiwan, China (*Shen et al., 2022*). A percentage change against the overall comparator period was calculated.

## Acknowledgements

This manuscript was prepared or accomplished by Jonine Figueroa in their personal capacity. The opinions expressed in this article are the author's own and do not reflect the view of the National Institutes of Health, the Department of Health and Human Services, or the United States government.

# Additional information

## Funding

| Funder | Grant reference number | Author |
|---|---|---|
| National Cancer Institute | Jonine Figueroa research is supported by the intramural program | Jonine Figueroa |

The funders had no role in study design, data collection and interpretation, or the decision to submit the work for publication.

## Author contributions

Reagan Lee, Conceptualization, Data curation, Investigation, Methodology, Writing – original draft, Writing – review and editing; Wei Xu, Data curation, Investigation, Methodology, Writing – review and editing; Marshall Dozier, Data curation, Methodology, Writing – review and editing; Ruth McQuillan, Conceptualization, Supervision, Methodology, Writing – review and editing; Evropi Theodoratou, Conceptualization, Supervision, Investigation, Methodology, Writing – review and editing; Jonine Figueroa, Data curation, Formal analysis, Supervision, Methodology, Writing – original draft, Writing – review and editing

## Author ORCIDs

Reagan Lee http://orcid.org/0000-0002-3658-4103
Wei Xu http://orcid.org/0009-0008-3338-4545
Marshall Dozier http://orcid.org/0000-0002-5151-1252
Ruth McQuillan http://orcid.org/0000-0003-0998-9540
Jonine Figueroa http://orcid.org/0000-0002-5100-623X

## Decision letter and Author response

Decision letter https://doi.org/10.7554/eLife.85680.sa1
Author response https://doi.org/10.7554/eLife.85680.sa2

# Additional files

## Supplementary files

• MDAR checklist

• Supplementary file 1. Full descriptive characteristics and data from included cross-sectional studies (n=26).

## Data availability

Source data included as *Supplementary file 1*.

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
