## [Editor Report]

This study presents important evidence of the impact of the covid pandemic on breast cancer screening globally but with important variations by healthcare setting. The data analysis is comprehensive, using solid systematic review methods. The results will be of interest to public health policymakers and health care and cancer control practitioners and researchers across the globe.

---

## [Decision Letter]

**Decision letter after peer review:**

Thank you for submitting your article "A Rapid review on the COVID-19 Pandemic's Global Impact on Breast Cancer Screening Participation Rates and Volumes from January-December 2020" for consideration by *eLife*. Your article has been reviewed by 2 peer reviewers, and the evaluation has been overseen by a Reviewing Editor and a Senior Editor. The following individual involved in the review of your submission has agreed to reveal their identity: Gwen Murphy (Reviewer #2).

Essential revisions:

1) Please see the suggestions from reviewer 2 about restructuring the discussion to improve clarity.

2) Please consider reviewer 2's suggestion about adding information on the type of lockdown in place in each of the regions/places/times covered by each study. This would definitely help the reader to understand the context of each study. I realize it might be difficult, but given its potential value added to the paper, if feasible, please consider adding it.

*Reviewer #2 (Recommendations for the authors):*

The authors have done a great job in gathering heterogeneous data from a range of sources and summarizing it. With such a range of data and further a range of contexts (different study settings and geographies), I think in some places the complexity is overwhelming the narrative. The paper would be stronger if you could determine what your most important points are and then discuss these in order of importance, particularly in the Discussion. I have some specific comments/questions below.

– The screening volume metric is described in the Methods as "volume…defined as total number of breast screening procedures" but then in the Tables it is described as % change. As a result, it's hard to interpret what the difference in the two metrics is, particularly in the few studies where both are used. Could you say more about this in the Discussion?

– In considering the impact of COVID, would it be worth adding a variable on what kind of "lock-down" was in place in that region during that time? This might not be included in all studies but should be readily google-able. I think this would be a strong addition because, as we know, restrictions differed dramatically across regions and countries and were likely the driving force behind the data.

– The Discussion is difficult to read. Thinking back to the restrictions in place in many countries during this period, many women might reasonably have been prevented from attending their screening because it was too far away, because they couldn't access transport, or because they had lost all childcare support. I don't think fear would necessarily be the reason for the lack of participation and, importantly, you have NO data to suggest this. I would begin your Discussion by restating your findings and then reminding people of the restrictions that were in place during this period and might explain your findings. Your narrative on screening capacity reduction is contradictory and could be seen as naïve – if there are reductions in staffing because of extraordinary constraints and staff illnesses then an information campaign to alleviate patient fears is unlikely to be effective.

Could you say more in your Discussion about the various study designs and how that might have contributed to the range of findings? Screening reportedly increased in one study but this appears to be a modelling study, what do you think of that?

The last paragraph of the Discussion is very well constructed and balanced in terms of repeating the main results and suggesting some possible explanations. I would look to reshape the earlier parts of the Discussion to mirror this structure and flow. You could also consider adding a line on the need to look at other screening programs (whether cancer or otherwise) and determine trends and implications there too.

---

## [Author Response]

Essential revisions:1) Please see the suggestions from reviewer 2 about restructuring the discussion to improve clarity.

We have restructured and edited the discussion as suggested.

2) Please consider reviewer 2's suggestion about adding information on the type of lockdown in place in each of the regions/places/times covered by each study. This would definitely help the reader to understand the context of each study. I realize it might be difficult, but given its potential value added to the paper, if feasible, please consider adding it.

We have added minimum and maximum infection rates within the study period for each country/region (WHO, 2023, UK Government, 2023; [18]; Government of Ontario, 2023; Government of the Netherlands, 2023; MDHHS, 2023; [37]; The Scottish Government, 2022). We did not include average infection rate as data was unavailable/uncertain (predicted/estimated/probable) for certain periods due to limited testing. We have also classified each lockdown restriction as per the Lancet paper (Li, Y. et al. (2020) ‘The Temporal Association of introducing and lifting non-pharmaceutical interventions with the time-varying reproduction number (R) of SARS-COV-2: A modelling study across 131 countries’,) and added them to Table 1. Bibliography and references have been updated accordingly.

Reviewer #2 (Recommendations for the authors):The authors have done a great job in gathering heterogeneous data from a range of sources and summarizing it. With such a range of data and further a range of contexts (different study settings and geographies), I think in some places the complexity is overwhelming the narrative. The paper would be stronger if you could determine what your most important points are and then discuss these in order of importance, particularly in the Discussion. I have some specific comments/questions below.

We thank the reviewer for the positive feedback and we have updated the discussion to improve clarity and readability.

– The screening volume metric is described in the Methods as "volume…defined as total number of breast screening procedures" but then in the Tables it is described as % change. As a result, it's hard to interpret what the difference in the two metrics is, particularly in the few studies where both are used. Could you say more about this in the Discussion?

We agree with the reviewer regarding the lack of clarity on the presentation of outcomes. We have attempted to standardise the definition of the outcome metrics in the Methods as we tended to use percentage changes in “volume” and “uptake rate” in presentation of our results.

– In considering the impact of COVID, would it be worth adding a variable on what kind of "lock-down" was in place in that region during that time? This might not be included in all studies but should be readily google-able. I think this would be a strong addition because, as we know, restrictions differed dramatically across regions and countries and were likely the driving force behind the data.

We have added minimum and maximum infection rates within the study period for each country/region (WHO, 2023, UK Government, 2023; Dipartimento della Protezione Civile. 2023; Government of Ontario, 2023; Government of the Netherlands, 2023; MDHHS, 2023; MCDHHS, 2023; The Scottish Government, 2022). We did not include average infection rate as data was unavailable/uncertain (predicted/estimated/probable) for certain periods due to limited testing. We have also classified each lockdown restriction as per the Lancet paper (Li, Y. et al. (2020) ‘The Temporal Association of introducing and lifting non-pharmaceutical interventions with the time-varying reproduction number (R) of SARS-COV-2: A modelling study across 131 countries’,) and added them to Table 1. Bibliography and references have been updated accordingly.

– The Discussion is difficult to read. Thinking back to the restrictions in place in many countries during this period, many women might reasonably have been prevented from attending their screening because it was too far away, because they couldn't access transport, or because they had lost all childcare support. I don't think fear would necessarily be the reason for the lack of participation and, importantly, you have NO data to suggest this. I would begin your Discussion by restating your findings and then reminding people of the restrictions that were in place during this period and might explain your findings. Your narrative on screening capacity reduction is contradictory and could be seen as naïve – if there are reductions in staffing because of extraordinary constraints and staff illnesses then an information campaign to alleviate patient fears is unlikely to be effective.

In rereading the discussion we agree with the reviewer that fear not a major driving force and have updated to include mitigation measures as measured in Li, Y 2020 as likely major reasons for change in screening participation and volume.

Could you say more in your Discussion about the various study designs and how that might have contributed to the range of findings? Screening reportedly increased in one study but this appears to be a modelling study, what do you think of that?

We are not clear about the comment on study designs as all were cross-sectional. The one study Nyante et al., was not a microsimulation type modelling study but rather used interrupted time series models which were are a quasi-experimental design used to evaluate the impact of interventions or exposures--in this case the exposure was in the setting of the COVID-19 pandemic and in the absence of the pandemic.

The last paragraph of the Discussion is very well constructed and balanced in terms of repeating the main results and suggesting some possible explanations. I would look to reshape the earlier parts of the Discussion to mirror this structure and flow. You could also consider adding a line on the need to look at other screening programs (whether cancer or otherwise) and determine trends and implications there too.

As suggested we have substantially revised the discussion to include more context and the data we incorporated on mitigation measures as suggested by the reviewer.